

# Evaluation of Mac-2 binding protein glycosylation isomer (M2BPGi) as a diagnostic marker for staging liver fibrosis: a meta-analysis

Siyao Gong, Xin Yu, Qian Li, Ming Chen, Shuguang Yu and Sha Yang

College of Acupuncture and Massage, Chengdu University of Traditional Chinese Medicine, Chengdu, China

## ABSTRACT

**Objective:** This study aimed to assess the accuracy of Mac-2 binding protein glycosylation isomer (M2BPGi) in predicting the stage of liver fibrosis.

**Methods:** Articles published until October 10, 2023, were searched in the PubMed, Embase, Web of Science, and Cochrane Library databases. Pooled sensitivity, specificity, diagnostic odds ratio (DOR), summary receiver–operator curves (SROC), and Spearman's rank correlation coefficient were used to examine the accuracy of M2BPGi in predicting the stage of liver fibrosis. A 95% confidence interval (CI) was provided for each estimate.

**Results:** Twenty-four studies were included in this meta-analysis, including 3,839 patients with liver fibrosis, 409 of whom progressed to stage 4 or above. The pooled sensitivity, specificity, and area under the ROC (AUC) for M2BPGi predicting liver fibrosis ≥F3 were 0.74 (95% CI [0.65–0.82]), 0.84 (95% CI [0.76–0.89]), and 14.99 (95% CI [9.28–24.21]), respectively. The pooled sensitivity, specificity, and AUC for ≥F4 were 0.80 (95% CI [0.70–0.88]), 0.80 (95% CI [0.73–0.86]), and 16.43 (95% CI [0.84–0.90]), respectively.

**Conclusion:** Among different sample partitions, M2BPGi has the best diagnostic performance for liver fibrosis stage ≥4. Furthermore, the cutoff of 1–2 is more accurate than that of 0–1 or 2–3 for fibrosis ≥ F3 and ≥ F4.

**Registration:** CRD42023483260.

## INTRODUCTION

Chronic liver disease is a progressive degeneration of liver functions for more than 6 months. The prevalence of the chronic liver disease is increasing with the aging of the population, excessive drinking, and obesity (*Berasain et al., 2023*; *Nasr et al., 2023*). In recent years, new cases of chronic hepatitis B, chronic hepatitis C, primary biliary cholangitis, and nonalcoholic fatty liver disease (NAFLD) have been constantly reported (*Nah et al., 2020*; *Shirabe et al., 2018*). Chronic hepatitis is a common disease induced by viral infection, which manifests as inflammation/necrosis of hepatocytes. The repeated regeneration of these cells will lead to liver fibrosis (*Ueda et al., 2020*). Patients with liver

Corresponding author
Sha Yang, yangsha@cdutcm.edu.cn

fibrosis often suffer from portal hypertension and hepatocellular carcinoma (HCC), which may increase their risk of mortality (*Tamai et al., 2021*). Therefore, early identification and specific immunotherapy of patients with liver fibrosis or liver cirrhosis are critical, which may regress liver fibrosis and liver cirrhosis, thereby preventing progression and decompensation (*Mak et al., 2019*). Accordingly, identifying the fibrosis stage in patients with chronic liver disease is of great importance.

Liver biopsy can help clinicians accurately determine the stage of fibrosis, but this invasive approach has various problems, such as inapplicability in clinical practice and frequent sampling errors. In addition, biopsy interpretation is subjective, and biopsy itself is associated with complications such as pain, serious bleeding, injury to other organs, and in rare cases, death (*Kanno et al., 2019*; *Choi et al., 2019*). Hence, it is necessary to develop a noninvasive method to predict the stage of fibrosis in patients with chronic liver disease. In recent years, multiple noninvasive methods, such as magnetic resonance imaging (MRI) and ultrasound elastography, have already been developed to assess the fibrosis stage. Despite the high accuracy of elastography in the diagnosis of hepatic fibrosis and hepatic steatosis, the equipment is not widely available due to high costs. As a result, there is an urgent need for a simple, cheap, accurate, and noninvasive mark to identify the stage of liver fibrosis (*Tamaki et al., 2021*).

Mac-2 binding protein glycosylation isomer (M2BPGi), a glycoprotein secreted by hepatic stellate cells during liver fibrosis progression (*Hur et al., 2022*), is widely recognized as a potential marker for liver fibrosis in NAFLD patients (*Mak et al., 2019*). Previous studies have suggested that the serum M2BPGi level can be used as a marker for the diagnosis of advanced fibrosis and liver cirrhosis in NAFLD patients (*Nah et al., 2020*; *Jang et al., 2021*). Nevertheless, some studies have reported different findings, and the diagnostic performance of M2BPGi remains unclear at each stage (*Ueda et al., 2020*). Thus, this study aimed to systematically assess the accuracy of M2BPGi in predicting the stage of liver fibrosis.

## METHODS

We performed a systematic review and meta-analysis following the Preferred Reporting Items for Systematic Reviews and Meta-Analyses (PRISMA) guidelines (the approval number for registration is CRD42023483260).

### Search strategy

Pubmed, Embase, Web of Science, and Cochrane Library databases were searched for relevant studies. Search terms included liver cirrhosis, Mac-2 binding protein glycosylation isomer, Mac-2 protein glycosylation isomer, and M2BPGi. The specific search strategy is provided in the Supplemental Materials.

### Inclusion and exclusion criteria

Inclusion criteria were as follows: Studies should have an explicit reference standard for the diagnosis of liver fibrosis (liver biopsy), report all or sufficient original data (*i.e.*, threshold

values, the number of true positive and false positive patients, sensitivity, and specificity, all of which are used to assess the accuracy of diagnosis). The following studies were excluded: animal experimental studies, secondary research, letters, case reports, conference summaries, guides, non-English studies, and those with unavailable full texts.

## Literature screening and data extraction

Retrieved studies were imported into Endnote X9.3.3 for management. After removing duplicate publications, the titles and abstracts were screened to exclude irrelevant studies. Then the full texts of the remaining studies were downloaded and read to determine the final included studies. Data were extracted from the included studies. Two independent investigators (Xin Yu and Qian Li) performed the literature screening and data extraction. Any disagreements were resolved by discussing with a third investigator (Siyao Gong).

## The quality assessment of the included studies

The quality of the included studies was independently evaluated by two investigators (Xin Yu and Qian Li) using the Quality Assessment of Diagnostic Accuracy Studies (QUADAS) tool (*Whiting et al., 2003*). The QUADAS tool encompasses two major domains: risk of bias and clinical applicability. The risk of bias domain consists of four questions: (1) Patient selection: could the selection of patients have introduced bias? (2) Index test: could the conduct or interpretation of the index test have introduced bias? (3) Reference standard: could the conduct or interpretation of reference standard have introduced bias? (4) Flow and timing: could the patient flow have introduced bias? The clinical applicability domain involves three concerns: (1) Patient selection: are there concerns that the included patients do not match the review question? (2) Index test: are there concerns that the conduct or interpretation of the index test differs from the review question? (3) Reference standard: are there concerns that the target condition as defined by the reference standard does not match the review question?

## Statistical analysis

Statistical analysis was performed using Meta-DiSc 14.0, Stata 15.1, and Revman 5.3. Specifically, Stata 15.1 was utilized to analyze various indicators and pool the effect sizes, including sensitivity, specificity, positive likelihood ratio, negative likelihood ratio, diagnostic odds ratio (DOR). DOR is a measure of the effectiveness of a diagnostic test. Estimated diagnostic accuracy measures were calculated using the $2 \times 2$ tables extracted from the included studies. Summary Receiver Operating Characteristic Curves (SROC) were plotted, and area under curve (AUC) was calculated. A 95% confidence interval (CI) was provided for each effect size. Assessment criterion was the Spearman correlation coefficient by Meta-DiSc 14.0, and a value of $P < 0.05$ indicated that no threshold impact had occurred. When $I^2 > 50\%$, the random-effects model was adopted for meta-analysis. Furthermore, Revman 5.3 was used to generate literature quality evaluation plots, and Deeks' funnel plot was used to assess the publication bias of the included studies. We have disclosed the data and stata code, as shown in Materials S1 and S2.

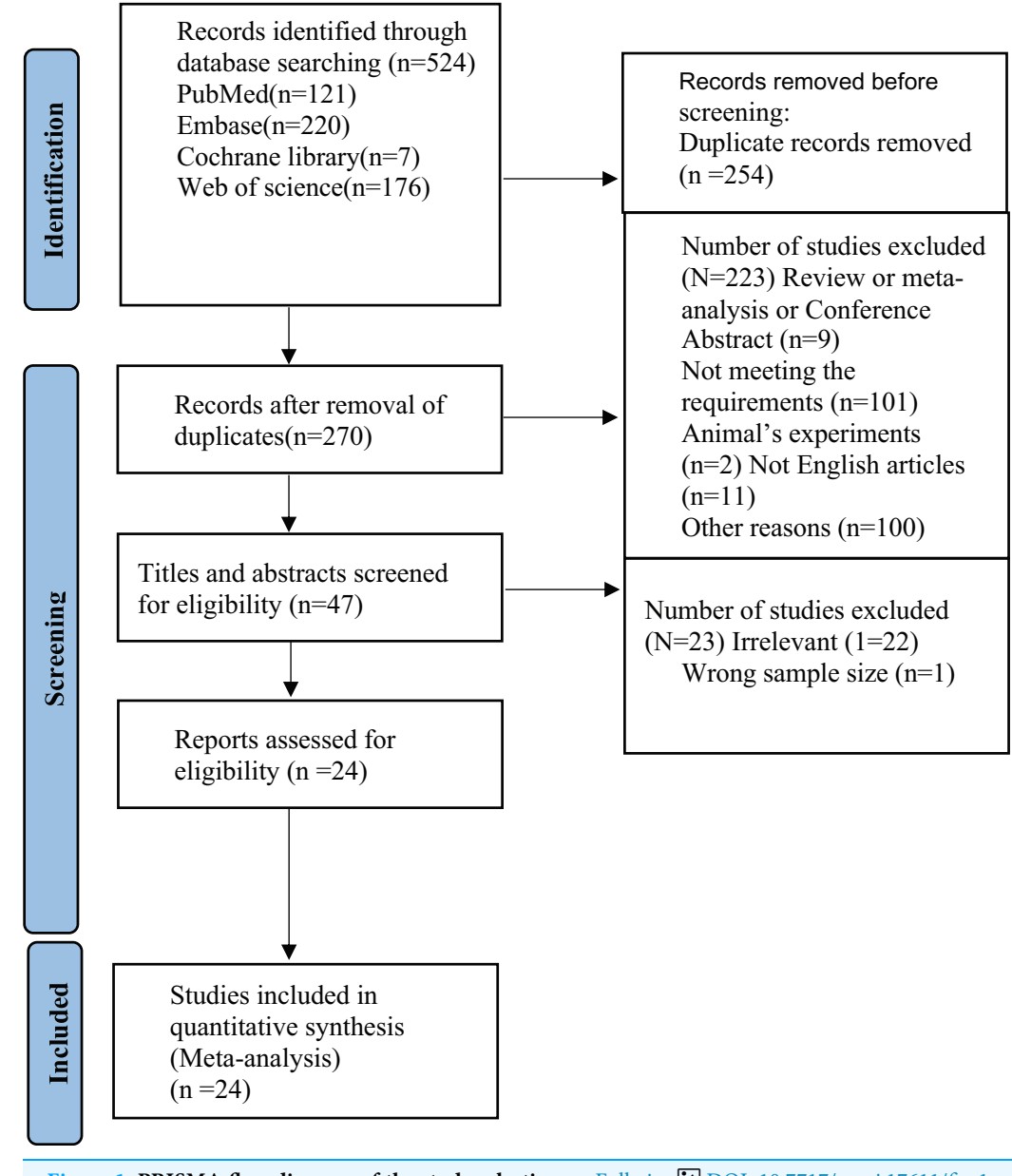

**Figure 1 PRISMA flow diagram of the study selection.**

## RESULTS

### Results of literature screening

A total of 527 studies were retrieved from the four aforesaid databases. After 270 duplicated articles were deleted, titles and abstracts were screened to exclude two animal experimental studies, nine reviews, and other irrelevant 196 articles. The full texts of the remaining studies were downloaded and read, and 26 studies were preliminarily eligible. However, 2 of the 26 trials were further excluded, one for wrong sample size and another for unavailable full text after contacting its authors. Ultimately, 24 trials were included in this study (*Nah et al., 2020*; *Ueda et al., 2020*; *Mak et al., 2019*; *Hur et al., 2022*;
**Table 1 Characteristics of the included studies.**

| Author | Year | Country | n | Age (years) | Sex (M/F) | BMI | ALT | AST |
|---|---|---|---|---|---|---|---|---|
| Yamada et al. | 2016 | Japan | 64 | 1.1 (0.4–16.0) | 16/48 | NA | 100 (9–83) | 158 (20–167) |
| Xu et al. | 2017 | China | 680 | 54.9 ± 11.0 | 465/215 | 23.6 ± 3.1 | 37.7 ± 43.8 | 49.2 ± 39.9 |
| Wei et al. | 2017 | China | 228 | 43 (18–71) | 157/71 | NA | 31 (7–75) | 31 (14–138) |
| Noguchi et al. | 2017 | Japan | 70 | 48.6 ± 14.1 | 37/33 | NA | 42.6 ± 39.4 | 35.8 ± 26.7 |
| Ueno et al. | 2018 | Japan | 37 | 3–38 | 12/25 | NA | 7–131 | 14–160 |
| Matsuura et al. | 2018 | Japan | 84 | 63 (54–69) | 48/36 | NA | 37 (25–58) | 38 (25–54) |
| Moon et al. | 2018 | Korea | 173 | 53.0 (44.0–73.0) | 105/68 | NA | NA | NA |
| Jekarl et al. | 2018 | Korea | 151 | 44.6 ± 12.6 | 101/50 | NA | NA | NA |
| Umetsu et al. | 2018 | Japan | 28 | 3–20 | 8/20 | NA | NA | NA |
| Mak et al. | 2019 | China | 240 | 47.5 | 116/124 | NA | 26 | 28 |
| Yamada et al. | 2019 | Japan | 116 | 11.8 (0.6–26.2) | 42/74 | NA | NA | NA |
| Nah et al. | 2020 | Korea | 236 | 27–81 | 176/60 | 17.1–34.1 | 9–184 | 0.35–4.14 |
| Tsuji et al. | 2020 | Japan | 96 | 51.1 ± 13.7 | 49/47 | NA | 46.8 ± 57.3 | 37.7±27.2 |
| Jang et al. | 2020 | Korea | 113 | 47.0 (30.0–61.0) | 58/55 | 28.7 (26.0–31.2) | 0.6–2.7 | 0.8–2.1 |
| Ueda et al. | 2020 | Japan | 108 | 69 (21–87) | 80/28 | 18.5–33.5 | 9–114 | 13–114 |
| Kim et al. | 2020 | Korea | 244 | 56.5 | 150/94 | NA | NA | NA |
| Numao et al. | 2020 | Japan | 141 | 67 | 66/75 | 23 | 43 | 43 |
| Kimura et al. | 2020 | Japan | 233 | 33 | 92/141 | NA | NA | NA |
| Cheng et al. | 2021 | China | 226 | 60.04 ± 10.08 | 103/123 | 25.28 ± 3.67 | 36.03 ± 21.96 | 24.84 ± 10.20 |
| Sasaki et al. | 2021 | Japan | 94 | 62.5 (28–78) | 53/41 | 23.6 (15.5–36.9) | 16.0 (6–203) | NA |
| Fujinaga et al. | 2021 | Japan | 102 | 61.0 ± 10.8 | 13/89 | NA | NA | NA |
| Jang et al. | 2021 | China | 80 | 46.4 | 51/29 | 29 | NA | NA |
| Hur et al. | 2022 | Korea | 152 | 50.2 ± 10.9 | 97/55 | NA | 19.0–41.8 | 23.0–34.0 |
| Mak et al. | 2022 | China | 143 | 58.7 | 101/42 | 23.3 | 28 | 27 |

*Jang et al., 2021*; *Xu et al., 2017*; *Matsuura et al., 2018*; *Ueno et al., 2018*; *Tsuji et al., 2020*; *Jekarl et al., 2018*; *Fujinaga et al., 2021*; *Wei et al., 2018*; *Mak et al., 2022*; *Kim et al., 2020*; *Noguchi et al., 2017*; *Cheng & Wang, 2021*; *Sasaki et al., 2021*; *Yamada et al., 2017*, *2019*; *Jang et al., 2022*; *Moon et al., 2018*; *Umetsu et al., 2018*; *Kimura et al., 2021*; *Numao et al., 2021*). The specific search strategy is presented in Fig. 1.

## Characteristics of the included studies

There were a total of 3,839 patients, aged from 0 to 87, of whom 2,196 (57.2%) were male. All included studies were retrospective. These patients were from China, Korea, and Japan. The basic characteristics of the study participants are shown in Table 1.

## Quality assessment

All included studies completed over 11 out of 14 items in the QUADAS list. This indicated that the overall quality of these studies was relatively high. The risk of bias assessment results are presented in Figs. 2 and 3.

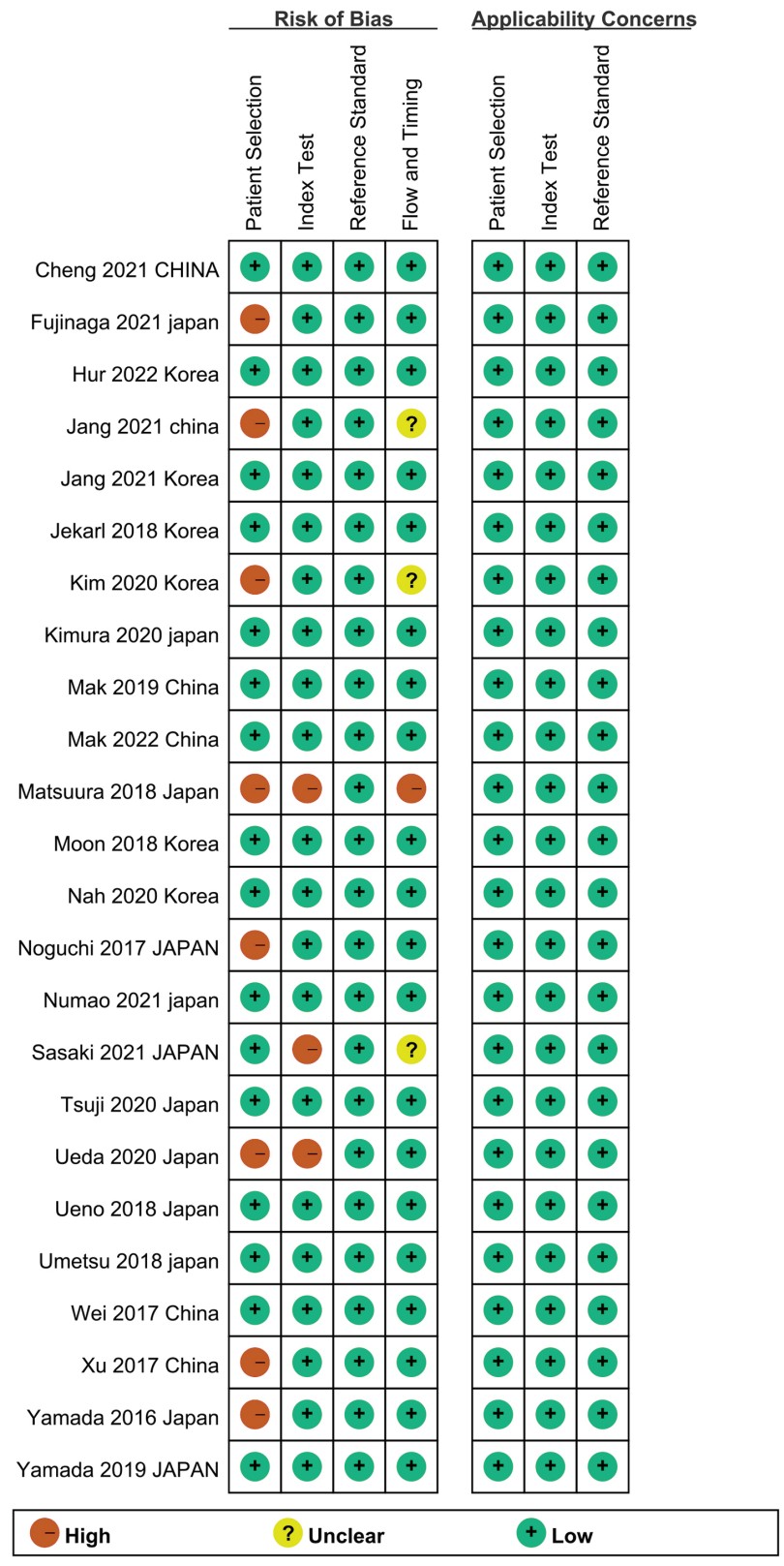

**Figure 2  Risk of bias.**

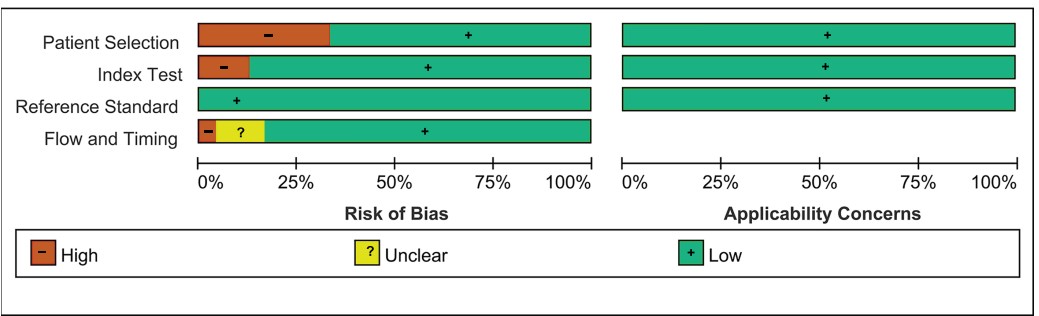

Figure 3 Summary of risk of bias.               

## Meta-analysis

Figure 4 shows the HSROC curves for the use of M2BPGi concentration for the diagnosis of the various stages of fibrosis in the participants. Table 2 shows the pooled results of the analysis of the use of M2BPGi concentration for the diagnosis of each fibrosis stage. We found that M2BPGi concentration offered good sensitivities and specificities for the identification of ≥F1, ≥F2, ≥F3, and ≥F4, with large AUCs.

M2BPGi diagnosis of ≥F1 had sensitivity and specificity values of 0.54 (95% CI [0.42–0.65]) and 0.81 (95% CI [0.63–0.91]), and the pooled DOR was 4.91 (95% CI [2.54–9.47]).

M2BPGi diagnosis of ≥F2 had sensitivity and specificity values of 0.71 (95% CI [0.63–0.79]) and 0.70 (95% CI [0.55–0.82]), and the pooled DOR was 5.80 (95% CI [3.55–9.49]).

M2BPGi diagnosis of ≥F3 had sensitivity and specificity values of 0.74 (95% CI [0.65–0.82]) and 0.84 (95% CI [0.76–0.89]), and the pooled DOR was 14.99 (95% CI [9.28–24.21]).

M2BPGi diagnosis of ≥F4 had sensitivity and specificity values of 0.80 (95% CI [0.70–0.88]) and 0.80 (95% CI [0.73–0.86]), and the pooled DOR was 16.43 (95% CI [0.84–0.90]).

## Publication bias

Publication bias was examined according to a Deeks' funnel plot generated by Stata. Due to $p$ value > 0.05 in all fibrosis stages, there was no publication bias (Fig. 5).

## DISCUSSION

To our knowledge, there is no systematic review assessing the accuracy of M2BPGi in the diagnosis of the stage of liver fibrosis. This study is the first systematic review that focuses on this topic. A total of 24 trials with 3,039 patients were included in this meta-analysis. The meta-analysis results revealed that the AUC values for fibrosis stages ≥1, ≥2, ≥3, and ≥4 were 0.69, 0.76, 0.86, and 0.87, respectively. This suggests that M2BPGi has the best diagnostic performance for fibrosis stage ≥4.

Liver biopsy remains the reference standard for the assessment of hepatic fibrosis, but it is invasive, difficult for repeated evaluation, and highly likely to have sampling error.

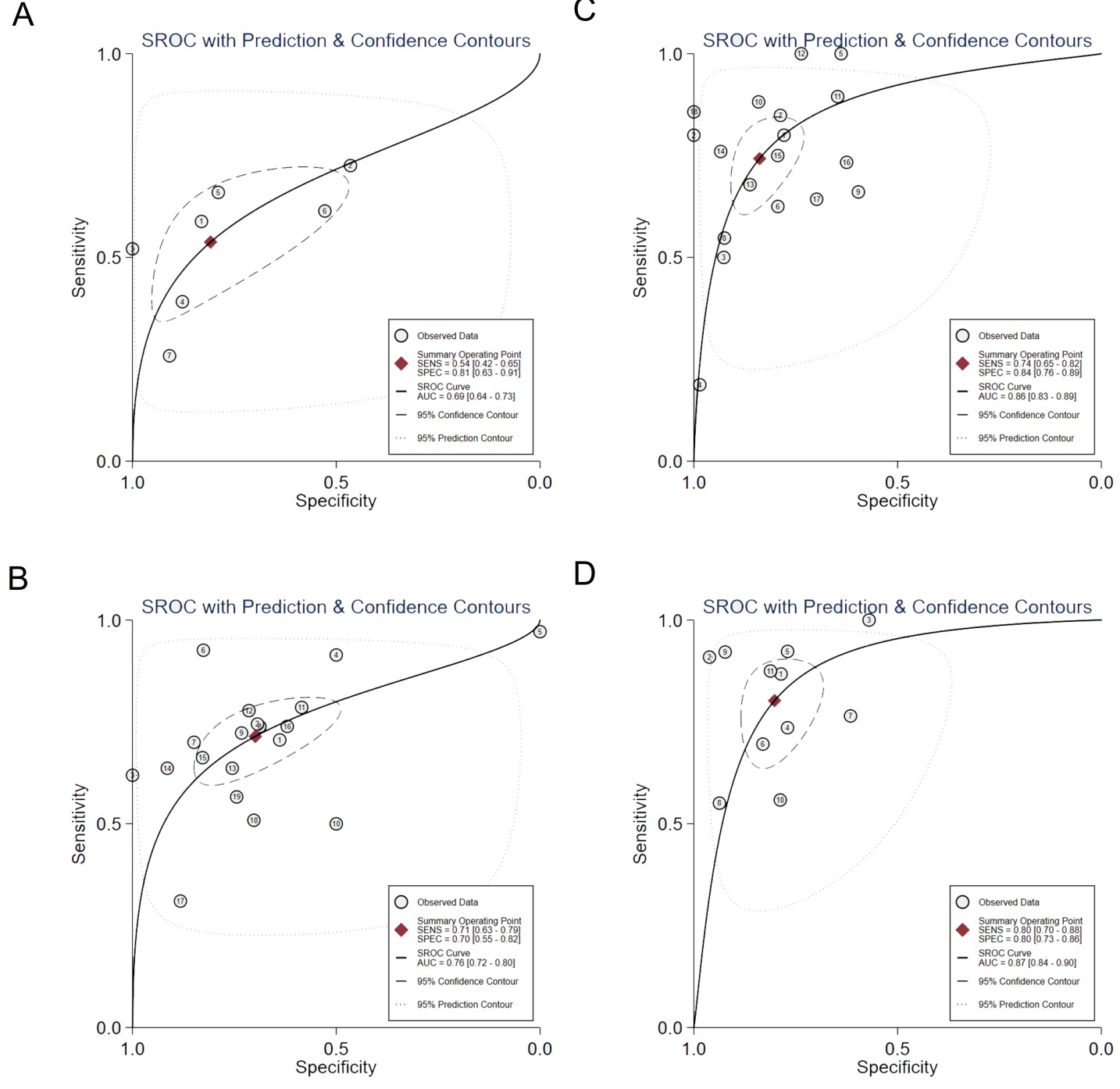

**Figure 4 Hierarchical summary receiver operating characteristic (SROC) curves for the use of M2BPGi concentration for the diagnosis of each stage of fibrosis (F1–F4).** (A) ≥F1. (B) ≥F2. (C) ≥F3. (D) ≥F4.

On top of this, this procedure cannot be applied frequently and is unacceptable for most patients (*Cheng & Wang, 2021*; *Sasaki et al., 2021*; *Noguchi et al., 2017*). M2BPGi, an isoform of the glycan structure of Mac-2 binding protein (M2BP), has been recently identified as a novel serum marker for liver fibrosis (*Nah et al., 2020*). It is an accurate and

**Table 2 Summary of the accuracy of the use of serum type IV collagen 7S concentration for the diagnosis of each stage of fibrosis.**

| Stage | Sensitivity (95%CI) | Specificity (95%CI) | Positive likelihood Ratio (95%CI) | Negative likelihood ratio (95%CI) | Diagnostic odds ratio (95%CI) | AUC (95%CI) | Deek's |
|---|---|---|---|---|---|---|---|
| F ≥ 1 | 0.54 [0.42–0.65] | 0.81 [0.63–0.91] | 2.81 [1.55–5.09] | 0.57 [0.48–0.68] | 4.91 [2.54–9.47] | 0.69 [0.64–0.73] | 0.11 |
| F ≥ 2 | 0.71 [0.63–0.79] | 0.70 [0.55–0.82] | 2.37 [1.61–3.50] | 0.41 [0.33–0.51] | 5.80 [3.55–9.49] | 0.76 [0.72–0.80] | 0.96 |
| F ≥ 3 | 0.74 [0.65–0.82] | 0.84 [0.76–0.89] | 4.60 [3.21–6.59] | 0.31 [0.22–0.42] | 14.99 [9.28–24.21] | 0.86 [0.83–0.89] | 0.57 |
| F ≥ 4 | 0.80 [0.70–0.88] | 0.80 [0.73–0.86] | 4.05 [2.91–5.63] | 0.25 [0.16–0.38] | 16.43 [8.97–30.08] | 0.87 [0.84–0.90] | 0.35 |

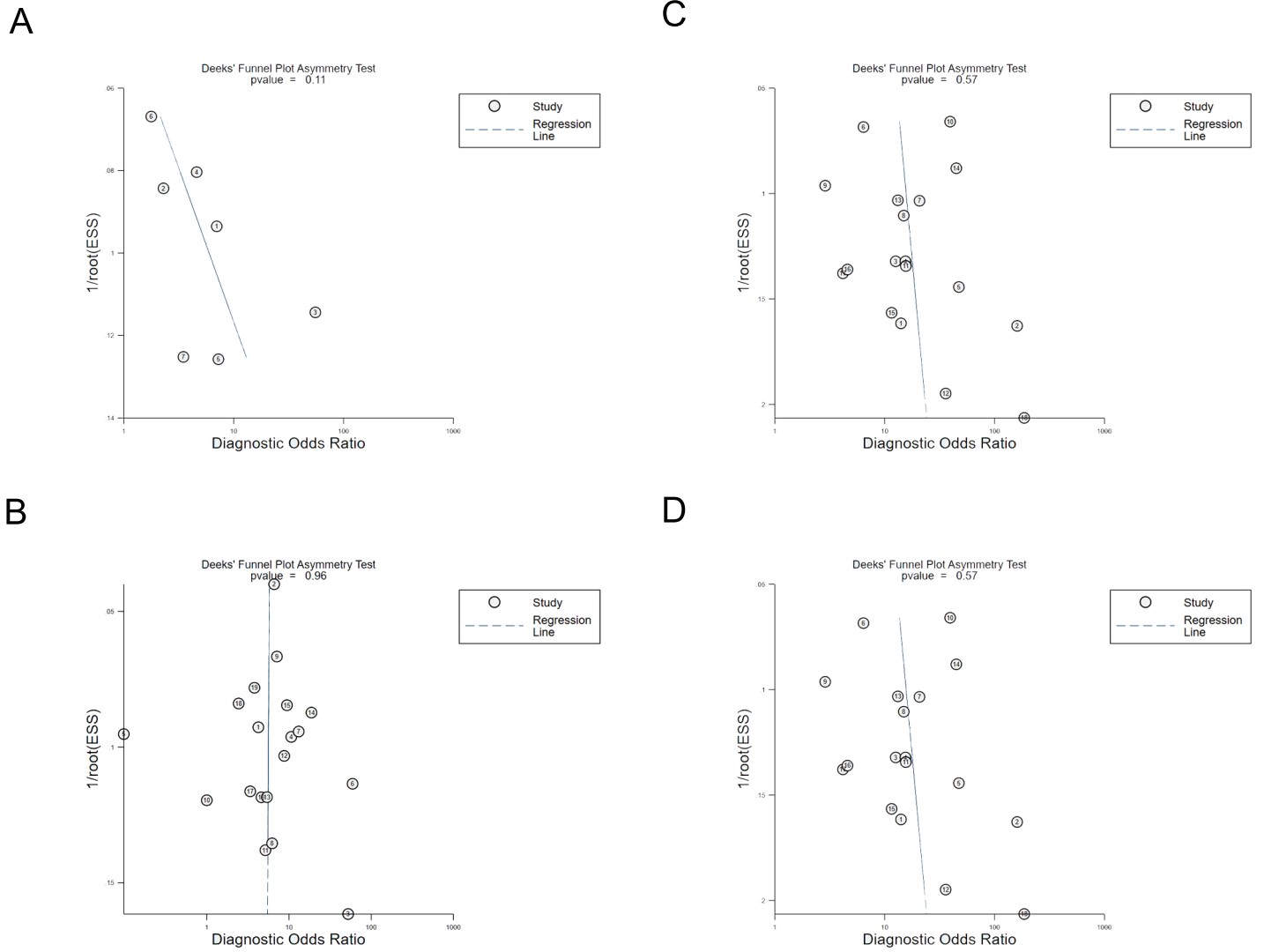

**Figure 5 Deeks' funnel plot used to assess publication bias for the use of M2BPGi concentration for the diagnosis of each stage of fibrosis (F1–F4).** (A) ≥F1. (B) ≥F2. (C) ≥F3. (D) ≥F4.

noninvasive approach to monitoring the degree of liver fibrosis (*Sasaki et al., 2021*). Previous studies, especially the studies on patients infected with HBV or HCV virus, have shown that M2BPGi, as a new reliable marker, has been used not only to predict the

**Table 3 Subgroup analyses for M2BPGi in the diagnosis of advanced fibrosis ≥F3 and ≥F4.**

| Cut off | Pooled sensitivity (95%CI) | Pooled specificity (95%CI) | AUROC | Pooled DOR (95%CI) |
|---|---|---|---|---|
| ≥F3 | | | | |
| 0–1 | 0.72 [0.65–0.78] | 0.77 [0.75–0.80] | 0.8341 | 9.98 [7.03–14.19] |
| 1–2 | 0.88 [0.83–0.92] | 0.77 [0.73–0.80] | 0.9053 | 25.99 [6.68–101.09] |
| 2–3 | 0.62 [0.54–0.69] | 0.90 [0.86–0.93] | 0.8025 | 12.71 [2.93–55.12] |
| ≥F4 | | | | |
| 0–1 | 0.78 [0.67–0.87] | 0.72 [0.67–0.76] | 0.8314 | 11.46 [6.24–21.05] |
| 1–2 | 0.67 [0.60–0.73] | 0.82 [0.79–0.84] | 0.8966 | 17.18 [6.08–48.56] |
| 2–3 | 0.78 [0.66–0.87] | 0.75 [0.69–0.80] | 0.8423 | 11.46 [4.31–30.46] |

prognosis of hepatopathy, the development of liver-related complications, and the risk of occurrence of HCC, but also to assess the degree of fibrosis in patients with viral hepatitis, biliary atresia, autoimmune hepatitis, and other fibrosis liver diseases (*Uojima et al., 2023*).

Subgroup analyses were conducted according to cutoff values at fibrosis stages ≥ 3 and ≥ 4. At fibrosis stage ≥ 3, the AUC was 0.8341 for the cutoff of 0–1, 0.9053 for the cutoff of 1–2, and 0.8025 for the cutoff of 2–3, indicating that the cutoff of 1–2 had the best diagnostic performance for fibrosis stage ≥ 3. At fibrosis stage ≥ 4, the AUC was 0.8314 for the cutoff of 0–1, 0.8966 for the cutoff of 1–2, and 0.8423 for the cutoff of 2–3, suggesting that the cutoff of 1–2 had the best diagnostic performance for fibrosis stage ≥ 4. In conclusion, the cutoff of 1–2 is the most accurate in the diagnosis of advanced hepatic fibrosis (fibrosis stage ≥ 3 or fibrosis stage ≥ 4). The results are shown in Table 3.

The present meta-analysis still has some limitations. Firstly, clinical applicability is restricted in that all included patients come from Asian countries, including China, Japan, and South Korea. Further studies are desired to include patients from more countries. Secondly, high heterogeneity in data leads to a failure to conduct subgroup analysis by country because all included patients come from Asian countries. At last, some included studies have a small sample size, which may affect the results.

## CONCLUSION

Among different sample partitions, M2BPGi is a non-invasive, accurate, reliable and acceptable method for predicting liver fibrosis stages. However, large-scale clinical studies in multiple countries are still needed to verify the results of this study.

## LIST OF ABBREVIATION

**HCC**        Hepatocellular carcinoma
**M2BPGi**    Mac-2 binding protein glycosylation isomer
**DOR**        Diagnostic odds ratio
**SROC**      Summary receiver–operator curves
**CI**          Confidence interval
**CLD**        Chronic liver disease

| NAFLD | Nonalcoholic fatty liver disease |
|---|---|
| MRI | Magnetic resonance imaging |
| M2BP | Mac-2 binding protein |

## ACKNOWLEDGEMENTS

We would like to thank the researchers and study participants for their contributions.

### Funding

This work was supported by the Innovation Team and Talents Cultivation Program of National Administration of Traditional Chinese Medicine (No: ZYYCXTD-D-202003), Central Guidance for Local Projects (NO. 2022ZYD0101), the National Natural Science Foundation of Sichuan Province (NO. 2023NSFSC0696), the National Natural Science Foundation of China (NO. 82004487), and the National Natural Science Foundation of China (Key Program) (NO. 82230127). The funders had no role in study design, data collection and analysis, decision to publish, or preparation of the manuscript.

### Grant Disclosures

The following grant information was disclosed by the authors:
Innovation Team and Talents Cultivation Program of National Administration of Traditional Chinese Medicine: ZYYCXTD-D-202003.
Central Guidance for Local Projects: 2022ZYD0101.
National Natural Science Foundation of Sichuan Province: 2023NSFSC0696.
National Natural Science Foundation of China: 82004487 and 82230127.

### Competing Interests

The authors declare that they have no competing interests.

### Author Contributions

- Siyao Gong conceived and designed the experiments, performed the experiments, analyzed the data, prepared figures and/or tables, authored or reviewed drafts of the article, and approved the final draft.
- Xin Yu conceived and designed the experiments, performed the experiments, analyzed the data, prepared figures and/or tables, authored or reviewed drafts of the article, and approved the final draft.
- Qian Li performed the experiments, analyzed the data, prepared figures and/or tables, authored or reviewed drafts of the article, and approved the final draft.
- Ming Chen conceived and designed the experiments, authored or reviewed drafts of the article, and approved the final draft.
- Shuguang Yu conceived and designed the experiments, authored or reviewed drafts of the article, and approved the final draft.

- Sha Yang conceived and designed the experiments, authored or reviewed drafts of the article, and approved the final draft.

## Data Availability

This is a systematic review/meta-analysis.

## Supplemental Information

Supplemental information for this article can be found online at http://dx.doi.org/10.7717/peerj.17611#supplemental-information.

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
