# Peer review of "Evaluation of Mac-2 binding protein glycosylation isomer (M2BPGi) as a diagnostic marker for staging liver fibrosis: a meta-analysis"

_PeerJ, doi:10.7717/peerj.17611_

## Round 0.1 · original submission · Minor Revisions

Dear authors,

We kindly request that you carefully review the comments provided by the reviewers. Their valuable suggestions offer insights to enhance your manuscript. Updating your work in accordance with their inputs will significantly strengthen its content.

Thank you

**Language Note:** The review process has identified that the English language must be improved. PeerJ can provide language editing services - please contact us at [email protected] for pricing (be sure to provide your manuscript number and title). Alternatively, you should make your own arrangements to improve the language quality and provide details in your response letter. – PeerJ Staff

Reviewer 1 ·

Basic reporting

For Figure 1, please export the figure in pdf/jpg format rather than using a screenshot. There are grammar error prompts left in the figure.
I suggest that the author should increase the resolution of the figures. The text can not be seen clearly.
I also suggest author provide the stata script and data to improve the reproducibility.

Experimental design

The analysis is too simple to support the conclusion claimed by the author.
I suggest authors do more analysis by splitting the dataset according to sample features such as age, gender, country, etc.

Validity of the findings

If the result is consistent across different sample partitions, the the author can make the claim. Otherwise, the result is not convincing.

Reviewer 2 ·

Basic reporting

Evaluation of Mac-2 binding protein glycosylation Isomer (M2BPGi) as a Diagnostic Marker for Staging Liver fibrosis: A Meta-Analysis

The manuscript adheres to the journal's guidelines by employing methods for data deidentification and upholding ethical standards. Furthermore, thorough scrutiny of the figures and tables has been undertaken. Overall, I commend the authors for their meticulous effort in presenting a meticulously structured research framework. The figures are pertinent and accurately labeled, facilitating a more lucid comprehension of the material.

However, the manuscript still requires revisions.
Provide reference for line 35-36.

Experimental design

Line 154, perhaps the authors meant P-value<0.05?
Provide references for line 168-172.
The statistical analysis section lacks depth and explanation of the statistical models built.
Can the authors please elaborate on how the statistical models were built, which were the predictors and the response variables to obtain the odds ratios reported in Tables 2 and 3?
Protein-Protein Interaction (PPI) could play a vital role in the biomarker’s performance, so have the author’s assessed any potential PPIs. The resource here could help: https://pharos.nih.gov/.
The statistical analysis lacks clarity and conciseness, making it difficult for readers to follow the description of the statistical analysis process. It could benefit from breaking down into smaller, more digestible chunks of information. The manuscript mentions the utilization of various statistical tools, but it doesn't provide sufficient explanation regarding why each tool was chosen or how they were applied in the analysis.

Validity of the findings

Overall, the manuscript could be improved by providing clearer explanations of the statistical analysis methods used, organizing the information more effectively, and ensuring that technical terms are explained for readers unfamiliar with the field.

---

## Round 0.2 · Minor Revisions

The authors should incorporate all comments suggested by the reviewers. Reviewer 1's comments were not addressed in the experimental design and findings in your manuscript. Update your manuscript according to the reviewer's comments. Thanks.

Reviewer 1 ·

Basic reporting

Please see additional comments.

Experimental design

Please see additional comments.

Validity of the findings

Please see additional comments.

Additional comments

Most problems have been improved but the features of each sample are not provided in the data and the Stata script that generates the result has still not been provided. I suggest the author add this information.

Reviewer 2 ·

Basic reporting

Authors have addressed the comments in my previous review. The article rerqires no further review and is ready for publication.

Experimental design

no comment

Validity of the findings

no comment

Additional comments

no comment

---

## Round 0.3 · Minor Revisions

The authors have conducted excellent work on this manuscript, incorporating feedback from reviewers to enhance the clarity and depth of the content. They have successfully elucidated the role of M2BPGi role in the Liver fibrosis, providing a detailed and comprehensive explanation of this complex biological process. Based on Reviewer 1 inputs, Please provide raw data and coding files for this study. Thanks

Reviewer 1 ·

Basic reporting

Please see additional comments.

Experimental design

Please see additional comments.

Validity of the findings

Please see additional comments.

Additional comments

The "raw data" is duplicated with "Additional material 1". They are both analysis results. Maybe the author uploaded it by mistake. No raw data is provided. And there is no code that is for loading data in the state code file.

---

## Round 0.4 · accepted · Accept

The authors have conducted excellent work on this manuscript, incorporating feedback from reviewers to enhance the clarity and depth of the content. They have successfully elucidated the role of Mac-2 Binding protein Glycosylation Isomer performed as a best diagnostic marker for liver fibrosis using meta analysis. This manuscript now offers a clearer and more insightful understanding of the mechanisms at play, thanks to the thoughtful integration of the reviewers' suggestions.

Reviewer 1 ·

Basic reporting

Please see additional comments.

Experimental design

Please see additional comments.

Validity of the findings

Please see additional comments.

Additional comments

The code has been provided.